# Free ISG15 Inhibits the Replication of Peste des Petits Ruminants Virus by Breaking the Interaction of Nucleoprotein and Phosphoprotein

Jingyu Tang,[a] Aoxing Tang,[a] Nannan Jia,[a] Hanyu Du,[a] Chuncao Liu,[a] Jie Zhu,[a] Chuanfeng Li,[a] Chunchun Meng,[a] Guangqing Liu[a]

aShanghai Veterinary Research Institute, Chinese Academy of Agricultural Sciences, Shanghai, People's Republic of China

**ABSTRACT** Peste des petits ruminants virus (PPRV) causes a highly contagious disease in small ruminants and severe economic losses in developing countries. PPRV infection can stimulate high levels of interferon (IFN) and many IFN-stimulated genes (ISGs), such as ISG15, which may play a key role in the process of viral infection. However, the role of ISG15 in PPRV infection and replication has not yet been reported. In this study, we found ISG15 expression to be significantly upregulated after PPRV infection of caprine endometrial epithelial cells (EECs), and ISG15 inhibits the proliferation of PPRV. Further analysis showed that free ISG15 could inhibit PPRV proliferation. Moreover, ISG15 does not affect the binding, entry, and transcription but does suppress the replication of PPRV. A detailed analysis revealed that ISG15 interacts and colocalizes with both viral N and P proteins and that its interactive regions are all located in the N-terminal domain. Further studies showed that ISG15 can competitively interact with N and P proteins and significantly interfere with their binding. Finally, through the construction of the C-terminal mutants of ISG15 with different lengths, it was found that amino acids (aa) 77 to 101 play a key role in inhibiting the binding of N and P proteins and that interaction with the P protein disappears after the deletion of 77 to 101 aa. The present study revealed a novel mechanism of ISG15 in disrupting the activity of the $N^0$-P complex to inhibit viral replication.

**IMPORTANCE** PPRV, a widespread and fatal disease of small ruminants, is one of the most devastating animal diseases in Africa, the Middle East, and Asia, causing severe economic losses. IFNs play an important role as a component of natural immunity against pathogens, yet the role of ISG15, an IFN-stimulated gene, in protecting against PPRV infection is currently unknown. We demonstrated, for the first time, that free ISG15 inhibits PPRV proliferation by disrupting the activity of the $N^0$-P complex, a finding that has not been reported in other viruses. Our results provide important insights that can further understand the pathogenesis and innate immune mechanisms of PPRV.

**KEYWORDS** peste des petits ruminants virus, free ISG15, $N^0$-P complex

Peste des petits ruminants (PPR) is a highly contagious viral disease with high morbidity and mortality rates of up to 100% and 90% in small ruminants, respectively (1). The rapid spread of PPR over the past few decades has become a concern for the Food and Agriculture Organization (FAO) of the United Nations and the World Organization for Animal Health (OIE), which have set a target for the global eradication of the disease by 2030. The disease is caused by peste des petits ruminants virus (PPRV), which belongs to the genus *Morbillivirus* in the family *Paramyxoviridae*. PPRV is an enveloped, single-stranded, negative-sense RNA virus whose 15,948-bp genome contains six genes encoding eight proteins, including nucleocapsid (N), phosphoprotein (P), matrix (M), fusion (F), hemagglutinin (H), and large (L) proteins, and two nonstructural

Address correspondence to Guangqing Liu, liugq@shvri.ac.cn, or Chunchun Meng, mengcc@shvri.ac.cn.

The authors declare no conflict of interest.

proteins, C and V, which are generated from the P gene through alternate start codons and RNA editing, respectively (2).

The PPRV genome is wrapped by N proteins to form a ribonucleoprotein (RNP) that serves as the template for transcription and replication (3). In the measles virus, the N protein contains two parts, folded Ncore and Ntail, an intrinsically disordered domain (4). The active P protein is a tetrameric structure whose oligomerization is required for efficient replication and transcription of RNA-dependent RNA polymerase (RdRp) (5). The P protein has a long disordered sequence at the N terminus, a tetrameric domain in the middle, and a folded domain (XD) at the C terminus, which is connected to the tetrameric domain by an unfolded loop (6). The N protein acts together with the P and L proteins to form the functional replication complex (7). In vesicular stomatitis virus (VSV), the N terminus of the P protein interacts with the central hinge region of RNA-free N ($N^0$) to form the $N^0$-P complex, which prevents the aggregation and nonspecific RNA binding of the N protein and plays a key role in regulating viral transcription and replication (8–10).

Interferon (IFN) plays an important role in the innate immune response and is an important line of defense against pathogens. Secreted IFN binds to its receptor and activates downstream signaling, ultimately inducing the expression of hundreds of IFN-stimulated genes (ISGs). Myxovirus resistance protein 1 (MX1), 2′,5′-oligoadenylate synthetase (OAS), viperin, tripartite motif protein 25 (TRIM25), ISG15, and many other ISGs have been reported to have antiviral effects (11–15). ISG15 plays an important role as one of the strongest and fastest induced ISG proteins. ISG15 was first identified in IFN-treated cells in 1979 by Farell et al. (16) and was the first member of the family of ubiquitin-like proteins (UBLs) to be identified. ISG15 has a molecular weight size of ~15 kDa, with a 17-kDa precursor protein that is subsequently processed by protease cleavage to the mature 15-kDa form (17). ISG15 uses its C-terminal LRLRGG motif to covalently bind to target proteins, thereby altering their biological functions, a process known as ISGylation. ISGylation is similar to ubiquitination and includes E1-activating enzyme, E2 binding enzyme, and E3 ligase. The cysteine activation site of the E1-activating enzyme forms an ATP-dependent thioester bond with the C terminus of ISG15, which is activated and transferred to the E2 binding enzyme by transesterification, consequently forming an ATP-dependent thioester bond between its C terminus and the cysteine activation site of the E2 binding enzyme. Under the action of the E3 linking enzyme, ISG15 uses the carboxyl group of the C-terminal two glycines to form an isopeptide bond with the target protein lysine, thus completing ISGylation (18–21). This process can be reversed by the ISG15-specific protease USP18, which removes ISG15 from the modified target protein (22).

Antiviral responses of ISGylation against a variety of viruses have been reported, such as vaccinia (23), influenza B (24), influenza A (25), Japanese encephalitis (26), classical swine fever (27), Zika fever (14), and human respiratory syncytial viruses (HRSVs) (28). An ISGylation-independent antiviral effect of ISG15 has also been demonstrated in hepatitis C (29), Chikungunya virus (30), and human cytomegalovirus (CMV) (31). However, the mechanisms of action of ISG15 against different viruses differ, and some remain unclear. It has been reported that ISG15 can inhibit different stages of viral infection, for example, entry (32), latency (33), replication (24, 28, 34), posttranslational modification and processing (35), egress and trafficking (36), assembly and budding (37), and release (38). ISGylation has also been reported to promote viral replication. ISGylation of RIG-I reduces the level of virus-induced IFN promoter activity and promotes the replication of the Newcastle disease virus (39). It has been observed that ISG15 acts differently on different viruses. However, the role of ISG15 in PPRV infection remains unclear.

It has been shown that ISG15 is expressed in goats and sheep vaccinated with PPRV, increases in goats and sheep infected with PPRV, and decreases with disease progression *in vitro*. In unvaccinated infected animals, ISG15 expression is higher than in vaccinated animals, which suggests a role for ISG15 in innate immune response (40).

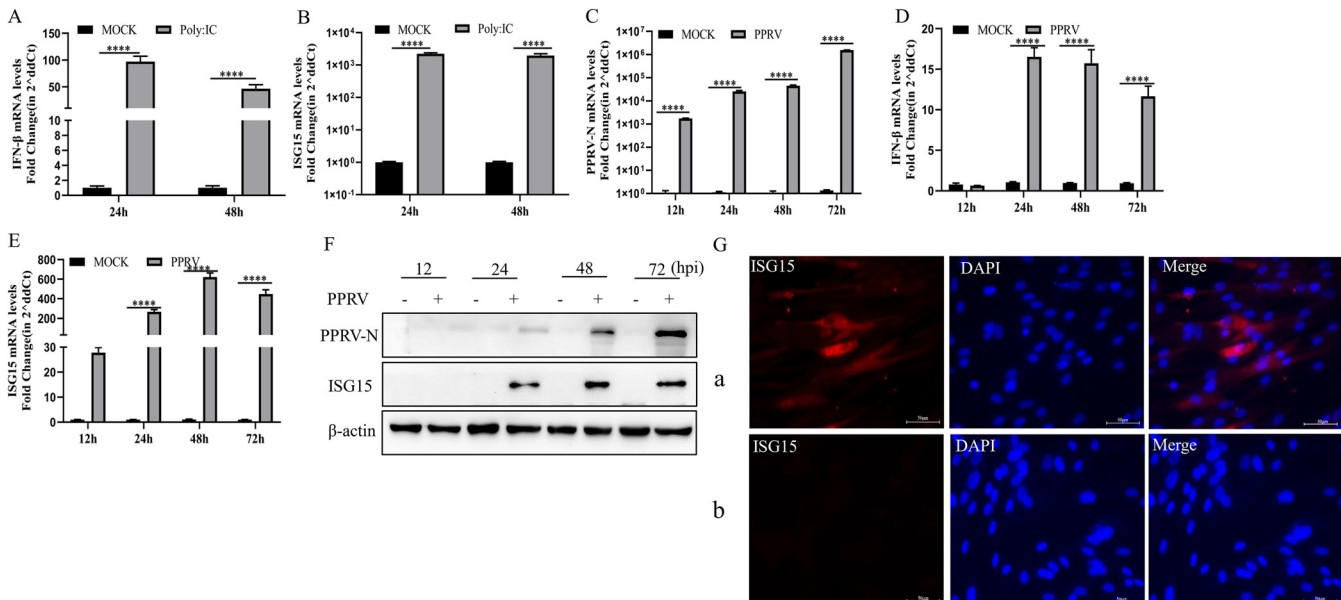

**FIG 1** Expression of ISG15 induced by PPRV infection of EECs. (A and B) EECs were treated with 2.5 $\mu$g/mL of poly(I·C) for 24 and 48 h, and levels of IFN-$\beta$ (A) and ISG15 (B) mRNA were measured by qRT-PCR. (C to E) EECs were infected with PPRV at an MOI of 1. Cells were harvested at 12, 24, 48, and 72 hpi, and levels of nucleoprotein (C), IFN-$\beta$ (D), and ISG15 (E) mRNA were measured by qRT-PCR. (F) EECs were infected with PPRV at an MOI of 1. Cells were harvested at 12, 24, 48, and 72 hpi and detected using Western blotting with anti-N, anti-ISG15, and anti-$\beta$-actin. (G) EECs were infected with PPRV at an MOI of 1. Cells were harvested at 36 hpi and detected using an immunofluorescence assay with anti-ISG15. The expression of ISG15 was detected after PPRV infected EECs. (G, a) PPRV-infected EECs; (G, b) uninfected EECs. Data were presented as means from three independent experiments and were statistically evaluated by two-way analysis of variance followed by Sidak's multiple-comparison tests. Statistically significant differences between groups are highlighted; ****, $P < 0.0001$.

Therefore, to explore whether ISG15 plays a role in the innate immune response induced by PPRV, we analyzed ISG15 expression during PPRV infection of caprine endometrial epithelial cells (EECs). Overexpression and knockout assays showed that ISG15 limited PPRV replication in EECs. We identified a novel mechanism by which ISG15 inhibits PPRV replication. This mechanism, which is not dependent on ISGylation, modulates the balance of the $N^0$-P complex to inhibit viral replication.

## RESULTS

**Expression of ISG15 induced by PPRV infection of EECs.** Previous studies have found that ISG15 can be stimulated by IFN-$\beta$ and viruses in different cells (14, 28). Polyinosinic poly(C) [Poly:IC], a synthetic analog of double-stranded RNA, was used to detect whether IFN-$\beta$ and ISG15 could be provoked in EECs. The results showed that Poly:IC strongly activated the transcriptional levels of IFN-$\beta$ and ISG15 in EECs at 24 and 48 h after transfection (Fig. 1A and B). To explore whether ISG15 expression is upregulated in PPRV infection *in vitro*, we investigated the expression of PPRV and ISG15 at both transcriptional and translational levels and IFN-$\beta$ at transcriptional levels following PPRV infection. The results showed that PPRV proliferated in EECs, with slow proliferation until 48 h, maximum proliferation at 72 h, and decreased proliferation thereafter (Fig. 1C). IFN-$\beta$ was not stimulated until after 12 h postinfection (hpi), after which it increased, reaching its maximum level at 24 hpi before decreasing (Fig. 1D). Figure 1E shows a large increase of ISG15 RNA in PPRV-infected EECs in a time-dependent manner. The increase started at 12 hpi and reached a maximum level at 48 hpi. ISG15 proteins were detected at 24 hpi and reached a maximum level at 48 hpi, after which they decreased (Fig. 1F). The expression of ISG15 was detected by indirect immunofluorescence assay in EECs infected with PPRV after 48 h. However, in EECs not infected with PPRV, the expression of ISG15 was not detected (Fig. 1G). Collectively, our results showed that PPRV infection can induce ISG15 expression in EECs.

**Knockdown and knockout of ISG15 enhanced PPRV replication.** To determine whether ISG15 has any anti-PPRV activity, EECs were transfected with control small

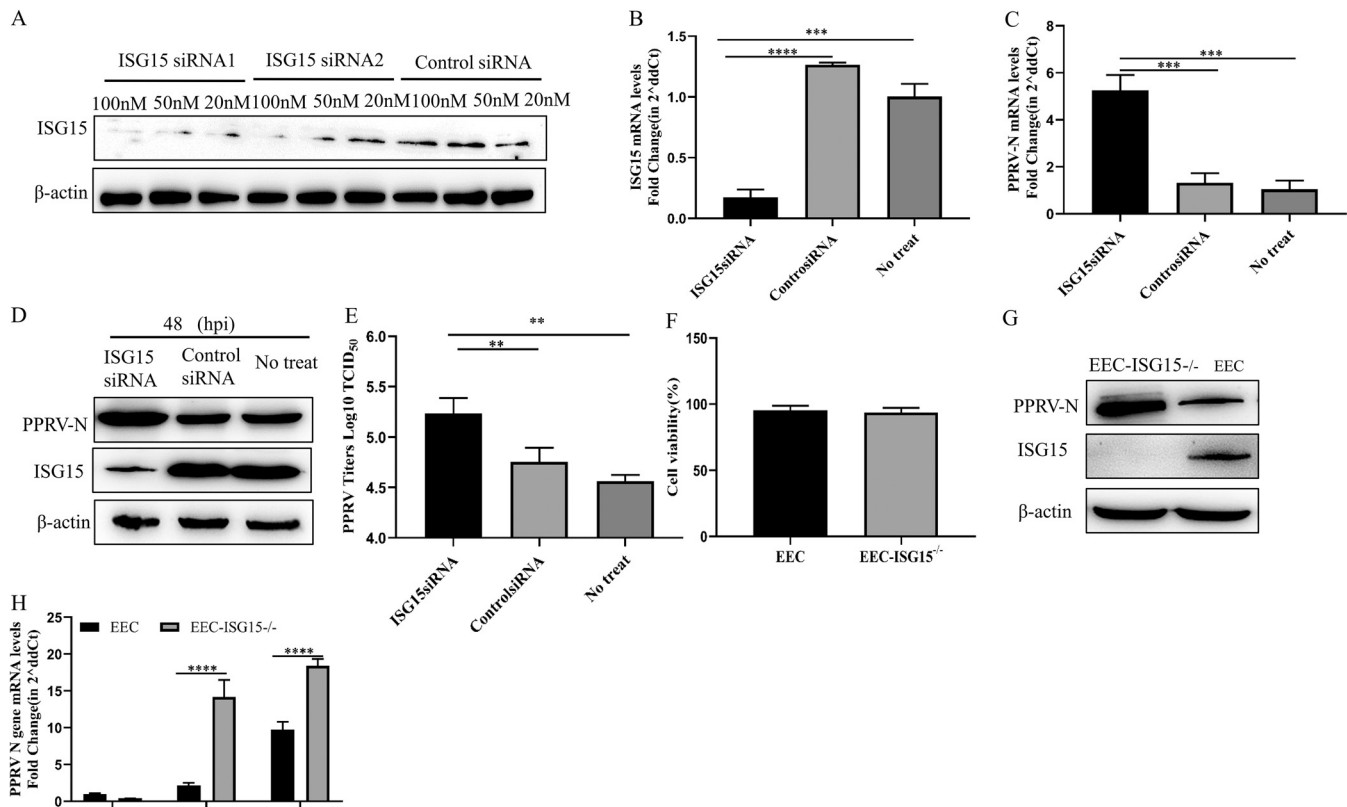

**FIG 2** Knockdown and knockout of ISG15 enhanced PPRV replication. (A) EECs were transfected with ISG15 siRNA1 (100 nM, 50 nM, and 20 nM), ISG15 siRNA2 (100 nM, 50 nM, and 20 nM), and control siRNA (100 nM, 50 nM, and 20 nM). Cells were harvested at 24 hpi and detected using Western blotting with anti-ISG15 and anti-$\beta$-actin. (B to E) EECs were transfected with ISG15 siRNA1 and control siRNA of 50 nM or not treated. At 24 h after transfection, cells were infected with PPRV at an MOI of 1. At 48 hpi, cells were harvested. Levels of ISG15 (B) and nucleoprotein (C) mRNA were measured by qRT-PCR. (D) Cells were detected using Western blotting with anti-N, anti-ISG15, and anti-$\beta$-actin. (E) Extracellular viral titers were quantified by virus titration assays and expressed as 50% tissue culture infective dose ($TCID_{50}$)/0.1 mL. (F) Viability of cell lines with a stable knockout of ISG15. (G) ISG15 knockout (EEC-ISG15$^{-/-}$) cells and EECs were infected with PPRV at an MOI of 1. Cells were harvested at 48 hpi and detected using Western blotting with anti-N, anti-ISG15, and anti-$\beta$-actin. (H) EEC-ISG15$^{-/-}$ cells and EECs were infected with PPRV at an MOI of 1. Cells were harvested at 24, 48, and 72 hpi, and the level of nucleoprotein mRNA was measured by qRT-PCR. Data were presented as means from three independent experiments and were statistically evaluated by one-way analysis of variance followed by Tukey's multiple-comparison test and two-way analysis of variance followed by Sidak's multiple-comparison tests. Statistically significant differences between groups are highlighted; **, $P < 0.01$; ***, $P < 0.001$; ****, $P < 0.0001$.

interfering RNA (siRNA) or specific ISG15 siRNA before being infected with PPRV. ISG15 expression in transfected ISG15 siRNA1 and siRNA2 cells was decreased compared with that in transfected control siRNA cells, and ISG15 siRNA1 had higher knockdown efficiency (Fig. 2A). EECs were transfected with ISG15 siRNA or control siRNA, and, after 24 h, they were infected with PPRV (multiplicity of infection [MOI] of 1). The transcriptional and translational levels and virus titers were measured following PPRV infection. As expected, a clear inhibition of ISG15 was observed in ISG15-silenced cells at both the transcriptional and translational levels at 48 h after PPRV infection (Fig. 2B and D). At the same time, ISG15-silenced cells revealed an increase in PPRV N protein transcriptional levels compared with control and untreated cells (Fig. 2C). A significant increase in N protein translational levels and virus titers at 48 h after PPRV infection in ISG15-silenced cells was observed compared with control and untreated cells at the same time postinfection (Fig. 2D and E).

To confirm the above-described results, ISG15$^{-/-}$ cells were constructed using CRISPR/Cas9 technology. The impact of ISG15 knockout on EEC death was assessed by 3-(4,5-dimethylthiazol-2-yl)-5-(3-carboxymethoxyphenyl)-2-(4-sulfophenyl)-2H-tetrazolium (MTS) assay, with no difference found between wild-type and ISG15 knockout cells (Fig. 2F). ISG15 expression in knockout cells was undetected, and PPRV N protein translational levels were elevated (Fig. 2G). The transcriptional level of PPRV N protein in EEC-ISG15$^{-/-}$ cells was significantly higher than that in EECs at 48 and 72 h after PPRV

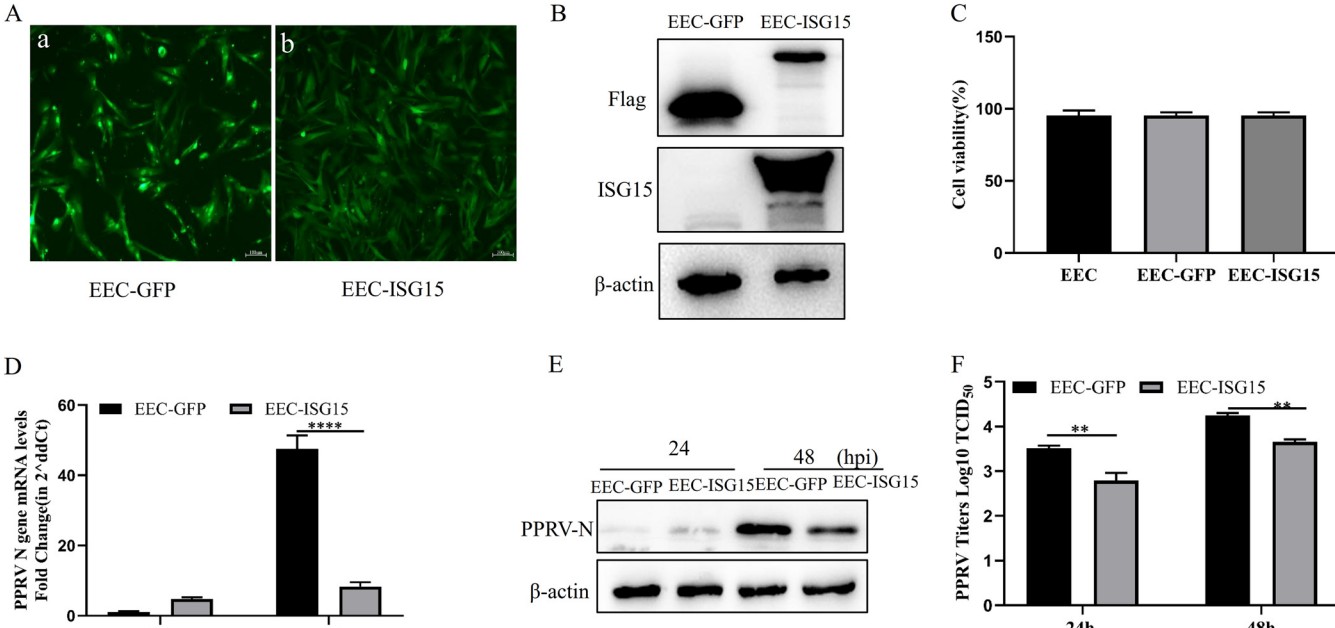

**FIG 3** Overexpression of ISG15 reduces PPRV replication in EECs. EECs stably overexpressing ISG15-GFP or GFP were constructed by lentiviral transduction. (A) GFP reporter expression was detected in EEC-GFP (a) and EEC-ISG15 (b). (B) Expression of ISG15 and GFP in EECs was determined by Western blotting with anti-Flag, anti-ISG15, and anti-$\beta$-actin. (C) Viability of cell lines stably overexpressing ISG15. (D to F) EEC-GFP and EEC-ISG15 cells were infected with PPRV at an MOI of 1. Cells were harvested at 24 and 48 hpi. (D) Levels of nucleoprotein mRNA were measured by qRT-PCR. (E) Cells were detected using Western blotting with anti-N and anti-$\beta$-actin. (F) Extracellular viral titers were quantified by virus titration assays and expressed as $TCID_{50}$/0.1 mL. Data were presented as means from three independent experiments and were statistically evaluated by two-way analysis of variance followed by Sidak's multiple-comparison tests. Statistically significant differences between groups are highlighted; **, $P < 0.01$; ****, $P < 0.0001$.

infection (Fig. 2H). These results demonstrate that the reduction of ISG15 can promote the proliferation of PPRV in EECs.

**Overexpression of ISG15 reduced PPRV replication.** To confirm the antiviral effect of ISG15 against PPRV, EEC cell lines expressing ISG15 (EEC-ISG15) and green fluorescent protein (EEC-GFP) were constructed. Green fluorescence was visualized in EEC-GFP and EEC-ISG15 under inverted fluorescence microscopy (Fig. 3A), and the overexpression of ISG15 in EEC-ISG15 and EEC-GFP was detected by anti-Flag and anti-ISG15 antibodies (Fig. 3B). Cell proliferation and viability were not affected (Fig. 3C). These results indicated that EEC cell lines expressing ISG15 and GFP were successfully constructed. The anti-PPRV effect of ISG15 was investigated in EEC-ISG15 cell line, and the transcriptional and translational levels and virus titers were measured at 24 and 48 h after PPRV infection. The results showed that the effect was not obvious compared with that in EEC-GFP at 24 hpi, but the PPRV N protein transcriptional levels were significantly inhibited compared with that in EEC-GFP (Fig. 3D). N protein translational levels revealed a decreased accumulation in EEC-ISG15 compared with control cells (Fig. 3E). PPRV titers in culture supernatants of EEC-ISG15 were significantly decreased compared with those in EEC-GFP at 24 and 48 hpi (Fig. 3F). These data suggest that the overexpression of ISG15 inhibits PPRV replication in EECs.

**The anti-PPRV activity of ISG15 is not dependent on ISGylation.** To determine in which form ISG15 exerts its anti-PPRV activity, we mutated the two glycines at the C terminus of ISG15 to alanines by point mutation. The amino acid mutation was proven to be successful by sequence determination (Fig. 4A). EECs were transfected with Myc-ISG15, ISG15AA, and Myc empty as control, followed by PPRV infection. The transcriptional and translational levels were measured following PPRV infection. The results showed that the transcriptional and translational levels of the N protein were significantly reduced in EECs transfected with ISG15 and ISG15AA compared with the controls. The reduction was more obvious in EECs transfected with ISG15 (Fig. 4B and C).

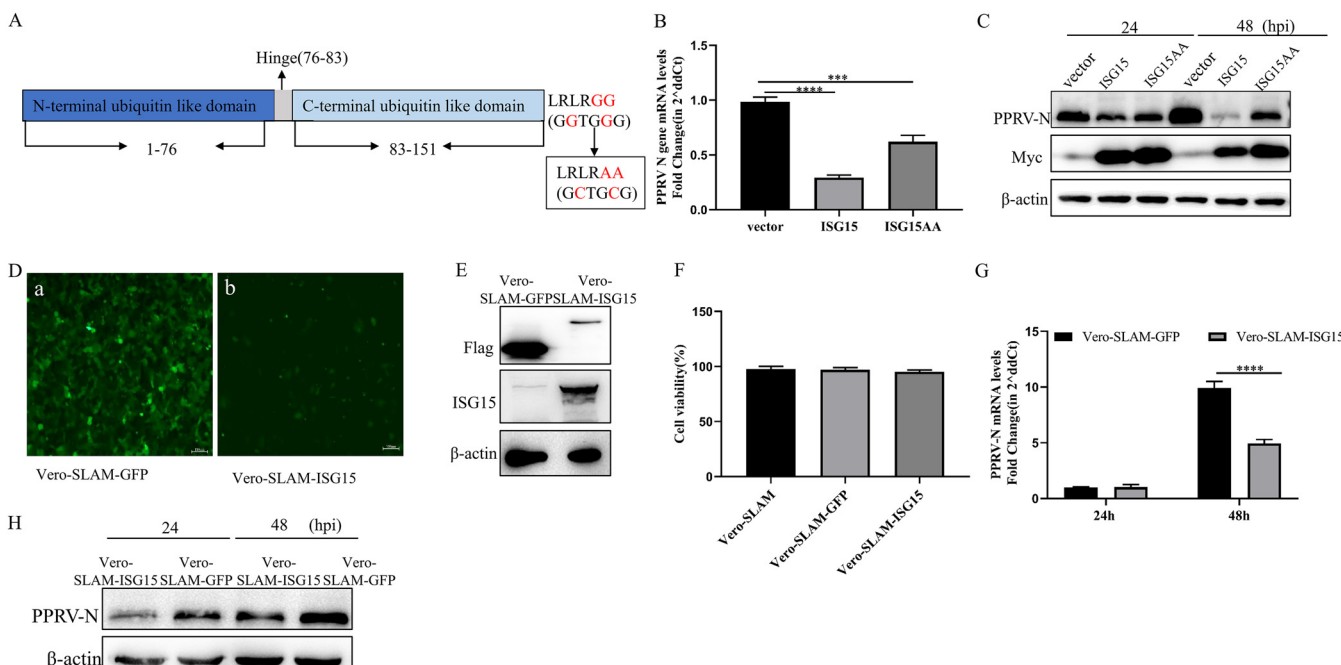

**FIG 4** Antiviral activity of ISG15 against PPRV is not dependent on ISGylation. (A) Construction of the ISG15AA mutant. (B) EECs were transfected with Myc-ISG15, Myc-ISG15AA, and vector. At 24 h after transfection, cells were infected with PPRV at an MOI of 1. At 48 hpi, cells were harvested. Nucleoprotein mRNA levels were measured by qRT-PCR. (C) EECs were transfected with Myc-ISG15, Myc-ISG15AA, and vector. At 24 h after transfection, cells were infected with PPRV at an MOI of 1. Cells were harvested at 24 and 48 hpi and detected using Western blotting with anti-N, anti-Myc, and anti-$\beta$-actin. (D) Vero-SLAM cells stably overexpressing ISG15-GFP or GFP were constructed by lentiviral transduction. GFP reporter expression was detected in Vero-SLAM-GFP (a) and Vero-SLAM-ISG15 (b). (E) The expression of ISG15 and the GFP protein in Vero-SLAM cells was determined by Western blotting using anti-Flag and anti-ISG15 antibodies, respectively. $\beta$-Actin served as an internal control. (F) Viability of cell lines stably overexpressing ISG15. (G to H) Vero-SLAM-GFP and Vero-SLAM-ISG15 cells were infected with PPRV at an MOI of 1. Cells were harvested at 24 and 48 hpi. (G) Levels of nucleoprotein mRNA were measured by qRT-PCR. (H) Cells were detected using Western blotting with anti-N and anti-$\beta$-actin. Data were presented as means from three independent experiments and were statistically evaluated by one-way analysis of variance followed by Tukey's multiple-comparison test and two-way analysis of variance followed by Sidak's multiple-comparison tests. Statistically significant differences between groups are highlighted; ***, $P < 0.001$; ****, $P < 0.0001$.

Vero cells are IFN secretion deficient, and, unlike other common mammalian cells, they cannot secrete IFN and ISGs, including ISGylation enzymes, when infected by a virus. Accordingly, we overexpressed ISG15 and an empty vector CMV as a control in Vero cells overexpressing the SLAM receptor of PPRV (Vero-SLAM cells). Green fluorescence was visualized in Vero-SLAM-GFP and Vero-SLAM-ISG15 under inverted fluorescence microscopy (Fig. 4D), and the overexpression of ISG15 in Vero-SLAM-GFP and Vero-SLAM-ISG15 was detected by anti-Flag and anti-ISG15 antibodies (Fig. 4E). Cell proliferation and viability were not different from that of the control cells (Fig. 4F). These results indicated that Vero-SLAM cell lines expressing ISG15 and GFP were successfully constructed. PPRV was inoculated into the overexpression cell line, and the transcriptional and translational levels were measured at 24 and 48 h following PPRV infection. The results showed that there was no significant difference in transcriptional levels, N protein translational levels were lower than those in the control group at 24 hpi, and both transcriptional and translational levels of N protein were significantly lower in Vero-SLAM-ISG15 cells than in the control group at 48 hpi (Fig. 4G and H). These results suggest that free ISG15 could inhibit the proliferation of PPRV.

**ISG15 specifically inhibits the PPRV replication phase.** To investigate in which phase of the PPRV replication cycle ISG15 is involved, we performed three experiments. First, PPRV N protein transcriptional and translational levels were measured in EEC-ISG15 and EEC-GFP cells after virus binding and entry. The results indicated that ISG15 did not affect PPRV binding and entry into EECs (Fig. 5A to D).

We investigated whether ISG15 was involved in the regulation of the PPRV replication phase by constructing the PPRV minigenome. This minigenome includes plasmids expressing the full length of viral N, P, and L proteins, as well as plasmids with the

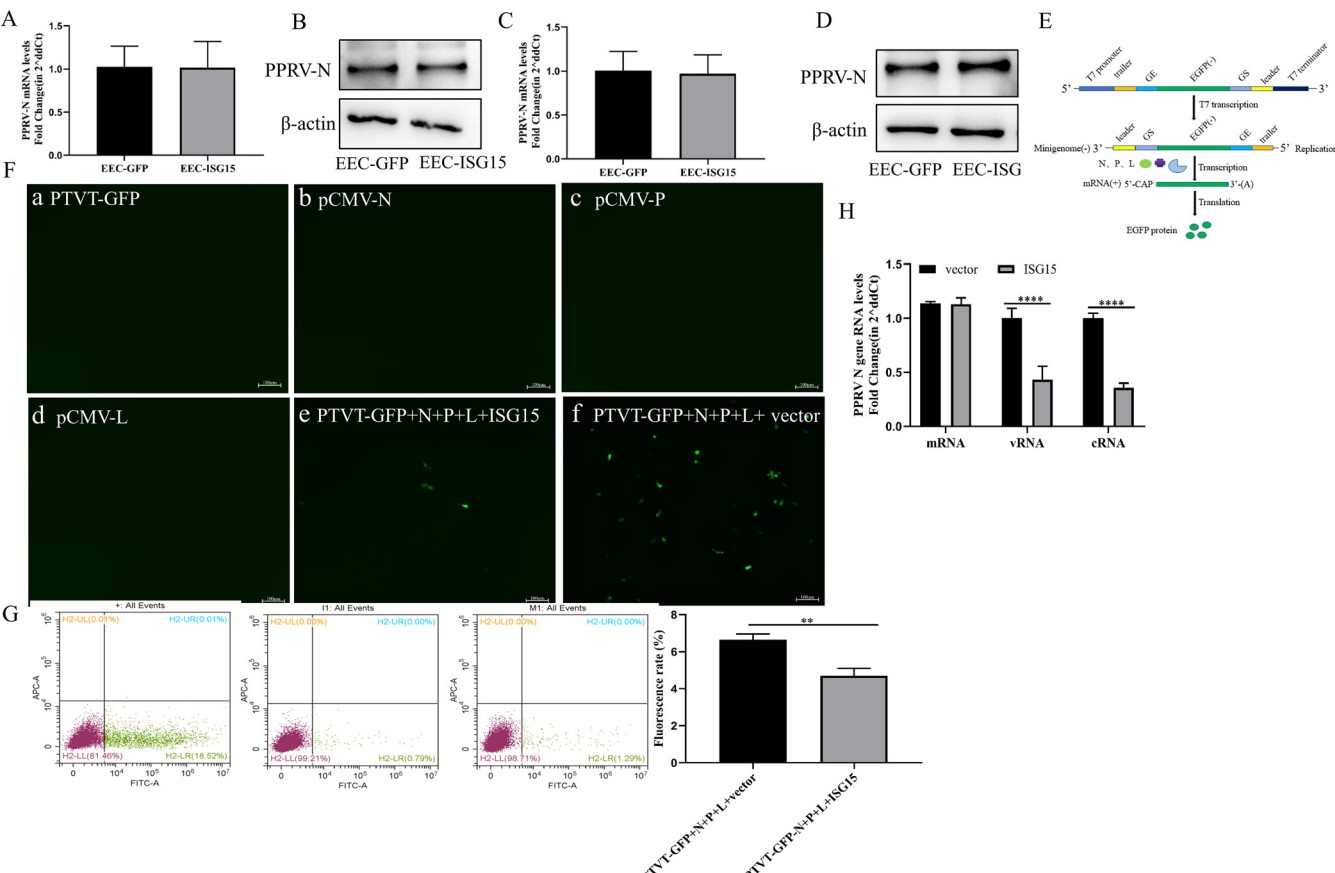

**FIG 5** ISG15 inhibits the PPRV replication phase. (A and B) EEC-ISG15 and EEC-GFP cells were incubated with PPRV at an MOI of 1 at 4°C for 1 h and then washed extensively with phosphate-buffered saline (PBS). Cell surface binding was assessed by determining the viral copy number in the cell lysates via qRT-PCR (A) and Western blotting with anti-N and anti-$\beta$-actin (B). (C and D) EEC-ISG15 and EEC-GFP cells were incubated with PPRV at an MOI of 1 at 4°C for 1 h and then allowed to internalize bound PPRV by incubation at 37°C for 10 min before an acid wash was used to remove cell surface virions. Viral entry into cells was assessed by determining the viral copy number in cell lysates by qRT-PCR (C) and Western blotting with anti-N and anti-$\beta$-actin (D). (E) Diagram of the minigenome structure. (F and G) BSR cells were transfected with PTVT-GFP, PCMV-N, PCMV-P, PCMV-L, and Myc-ISG15 or vector. (F) Cells were observed at 36 h using a fluorescence microscope. (G) The number of fluorescent cells was analyzed by flow cytometric analysis. The percentage of cells with green fluorescence in the positive control group was used as 100% to unify the percentage of green fluorescence in experimental groups. The percentage of green fluorescent cells in experimental groups was represented by a histogram. (H) Vero-SLAM cells were transfected with Myc-ISG15 and vector. At 24 h after transfection, cells were infected with PPRV at an MOI of 1. At 12 hpi, cells were harvested. Viral mRNA, vRNA, and cRNA were reverse transcribed by specific primers. Levels of three viral RNA species were measured by qRT-PCR. Data were presented as means from three independent experiments and were statistically evaluated by two-way analysis of variance followed by Sidak's multiple-comparison tests. Statistically significant differences between groups are highlighted; **, $P < 0.01$; ****, $P < 0.0001$.

reverse insertion of GFP such that GFP can reflect the complex activity of viral replication (Fig. 5E). BSR cells were transfected with the viral minigenome and ISG15 or an empty plasmid. Fluorescence in transfected ISG15 cells was lower than that in the control cells as seen by microscopy after 36 h (Fig. 5F). The level of fluorescence in transfected ISG15 cells was significantly lower than that in the control group by flow cytometric analysis (Fig. 5G). These data suggest that ISG15 inhibits the transcriptional replication phase of PPRV. Then, the mRNA, viral RNA (vRNA), and cRNA levels of PPRV were detected 12 h after PPRV infection and the overexpression of ISG15. It was found that the levels of vRNA and cRNA of PPRV were significantly lower than those of the control group (Fig. 5H), indicating that ISG15 affected the replication process of PPRV.

**ISG15 interacts with both N and P proteins.** To explore the mechanism by which ISG15 inhibits the replication phase, we tested potential interactions between ISG15 and viral replication-related proteins. HEK-293T cells were cotransfected with Myc-ISG15 and PPRV proteins with a Flag tag. The immunoprecipitation results showed that ISG15 interacted with N and P proteins (Fig. 6A and B) and did not interact with M and H proteins (Fig. 6C). *In vitro* binding assays showed that ISG15 interacted directly with

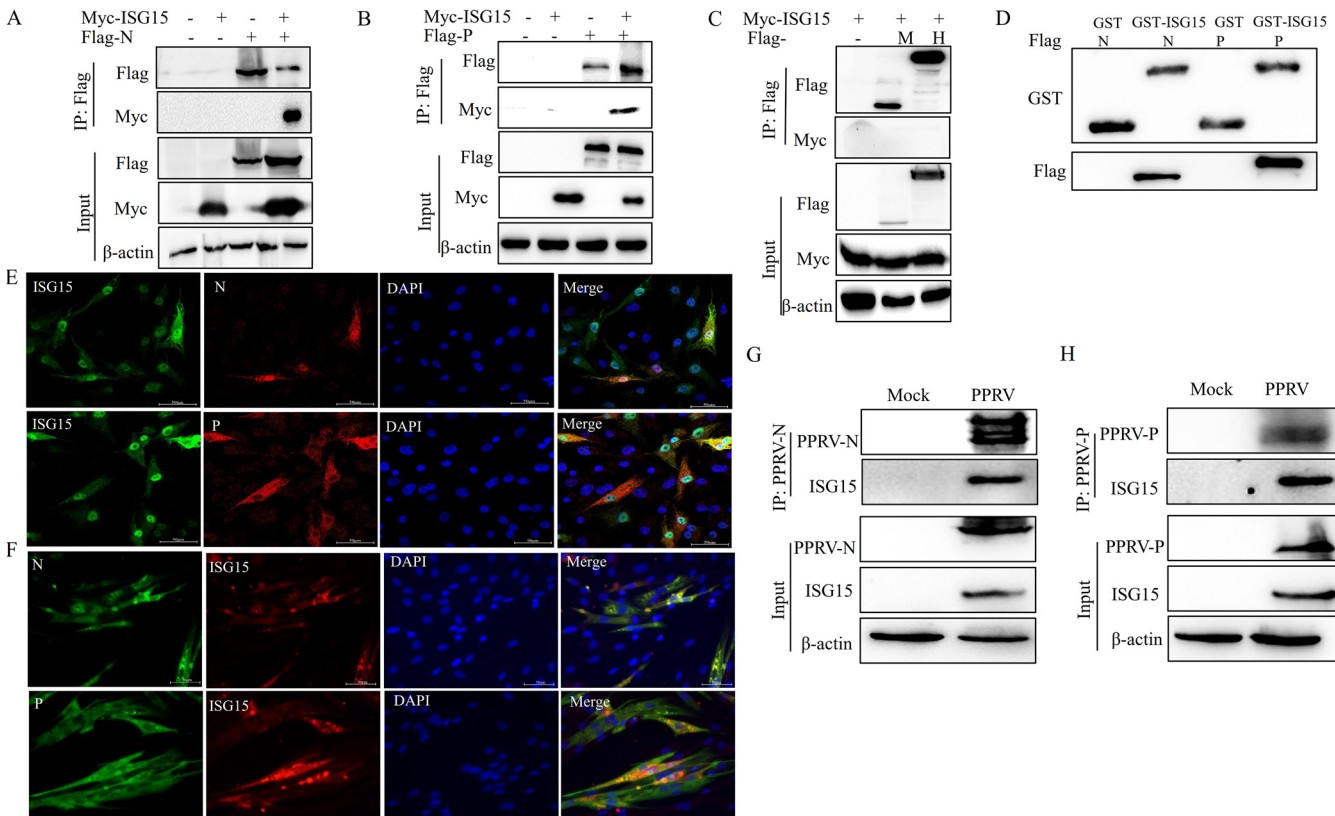

**FIG 6** ISG15 interacts with N and P proteins. (A) HEK-293T cells were cotransfected with Myc-ISG15 and Flag-N or vector. At 24 hours posttransfection (hpt), cells were harvested and immunoprecipitated with anti-Flag antibody. Immunoblot analysis was performed with anti-Myc and anti-Flag antibodies. Expression levels of the proteins were analyzed by immunoblot analysis of the lysates with anti-Flag, anti-Myc, and anti-$\beta$-actin antibodies. (B) HEK-293T cells were cotransfected with Myc-ISG15 and Flag-P or vector. At 24 hpt, cells were harvested and immunoprecipitated with anti-Flag antibody. Immunoblot analysis was performed with anti-Myc and anti-Flag antibodies. Expression levels of the proteins were analyzed by immunoblot analysis of the lysates with anti-Flag, anti-Myc, and anti-$\beta$-actin antibodies. (C) HEK-293T cells were cotransfected with Myc-ISG15 and Flag-M, Flag-H, or vector. At 24 hpt, cells were harvested and immunoprecipitated with anti-Flag antibody. Immunoblot analysis was performed with anti-Myc or anti-Flag antibodies. Expression levels of the proteins were analyzed by immunoblot analysis of the lysates with anti-Flag, anti-Myc, and anti-$\beta$-actin antibodies. (D) The GST fusion protein of ISG15 was purified on glutathione beads and incubated with lysates of HEK-293T cells transfected with Flag-N or Flag-P and analyzed by immunoblotting with an anti-Flag antibody. The GST protein was used as a negative control. (E) EECs were cotransfected with Myc-ISG15 and Flag-N or Flag-P. At 24 hpt, cells were harvested and detected using an indirect immunofluorescence assay with anti-Flag and anti-Myc antibodies. (F) EECs were infected with PPRV at an MOI of 1. Cells were harvested at 36 hpi and detected using an indirect immunofluorescence assay with anti-ISG15 and anti-N or anti-P antibodies. (G and H) EECs were infected with PPRV at an MOI of 1. Cells were harvested at 48 hpi and immunoprecipitated with anti-N (G) or anti-P (H) antibodies and further detected using immunoblot analysis with anti-N (or anti-P) and anti-ISG15 antibodies. Expression levels of the proteins were analyzed by immunoblot analysis of the lysates with anti-N (or anti-P), anti-ISG15, and anti-$\beta$-actin antibodies.

N and P proteins (Fig. 6D). Subsequently, HEK-293T cells were cotransfected with Myc-ISG15 and Flag-N or Flag-P, and then, colocalization experiments were performed. The results showed that ISG15 was distributed in the nucleus and cytoplasm, and it colocalized with the N protein in the nucleus and cytoplasm and with the P protein in the cytoplasm (Fig. 6E). To confirm that ISG15 interacted with replication complex subunit proteins during PPRV infection, an indirect immunofluorescence assay and immunoprecipitation were performed after PPRV infection with EECs. It was found that ISG15 was not detected in EECs without PPRV infection, but ISG15 was detected and colocated with viral N and P proteins after PPRV infection (Fig. 6F). Endogenous ISG15 interacts with N and P proteins of PPRV (Fig. 6G and H). These results suggest that ISG15 interacts with the N and P proteins of PPRV.

**ISG15 interferes with the interaction between N and P proteins.** To identify the key regions where ISG15 interacts with N and P proteins, we divided the PPRV N and P proteins into two segments based on their predicted structures (Fig. 7A). Since the specific region of interaction between N and P proteins in PPRV has not yet been reported, we first determined this region. Glutathione *S*-transferase-N (GST-N) and GST-P were used as bait proteins in glutathione pulldown assays. GST-N is bound to the P protein

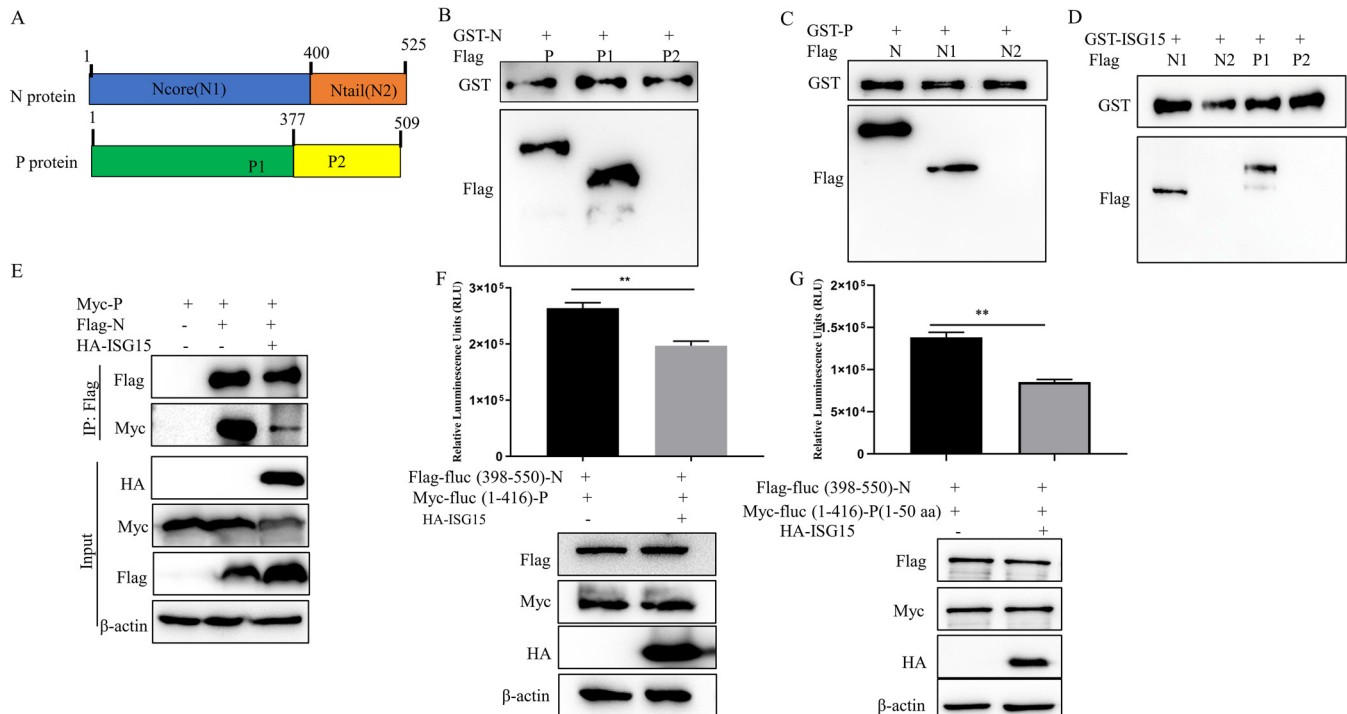

**FIG 7** ISG15 interferes with the interaction between N and P proteins. (A) Schematic representation of the strategy for constructing truncated N and P proteins. (B) The GST fusion protein of N was purified on glutathione beads and incubated with lysates of HEK-293T cells transfected with Flag-P, Flag-P1, or Flag-P2, and then analyzed by immunoblotting with an anti-Flag and anti-GST antibody. (C) The GST fusion protein of P was purified on glutathione beads and incubated with lysates of HEK-293T cells transfected with Flag-N, Flag-N1, or Flag-N2, and then analyzed by immunoblotting with an anti-Flag and anti-GST antibody. (D) The GST fusion protein of ISG15 was purified on glutathione beads and incubated with lysates of HEK-293T cells transfected with Flag-N1, Flag-N2, Flag-P1, or Flag-P2, and then analyzed by immunoblotting with an anti-Flag and anti-GST antibody. (E) HEK-293T cells were cotransfected with Flag-N, Myc-P, and HA-ISG15 or vector. At 24 hpt, cells were harvested and immunoprecipitated with anti-Flag antibody. Immunoblot analysis was performed with anti-Myc or anti-Flag antibodies. Expression levels of the proteins were analyzed by immunoblot analysis of the lysates with anti-Flag, anti-Myc, anti-HA, and anti-$\beta$-actin antibodies. (F) According to the luciferase complementary system, HEK-293T cells were cotransfected with Flag-fluc (aa 398 to 550)-N, Myc-fluc (aa 1 to 416)-P, and HA empty or HA-ISG15. Cell lysates were collected after 24 h to detect luciferase intensity. (G) According to the luciferase complementary system, HEK-293T cells were cotransfected with Flag-fluc (aa 398 to 550)-N, Myc-fluc (aa 1 to 416)-P (aa 1 to 50), and HA empty or HA-ISG15. Cell lysates were collected after 24 h to detect luciferase intensity. Data were presented as means from three independent experiments and were statistically evaluated by Student's $t$ test. **, $P < 0.01$.

and its N terminus (Fig. 7B). Similarly, GST-P is bound to the N protein and its N terminus (Fig. 7C). Then, GST-ISG15 was used as a bait protein in the glutathione pulldown assays, and the results showed that GST-ISG15 was bound to the N termini of both the N and P proteins (Fig. 7D). As the region where ISG15 interacts with N and P proteins is consistent with the region where N and P proteins interact, we speculated that the binding of ISG15 to N and P proteins would affect the interactions between these proteins. To test this hypothesis, HEK-293T cells were cotransfected with Flag-N, Myc-P, hemagglutinin (HA)-ISG15, Flag-N, and Myc-P as a control. The immunoprecipitation results showed that the interaction between N and P proteins was weakened after transfection with ISG15 compared with the control (Fig. 7E). According to the luciferase complementary system constructed in the laboratory previously, two segments of luciferase were connected to the N protein and P protein of PPRV, respectively, namely, Flag-fluc (398 to 550)-N and Myc-fluc (1-416)-P. HEK-293T cells were cotransfected with the two plasmids and HA empty or HA-ISG15, respectively. The luciferase results showed that the signal intensity of transfected ISG15 was significantly lower than that of the control group (Fig. 7F). The above results indicated that ISG15 competitively bound the N protein and P protein, which weakened their interaction. Previous studies have shown that P (amino acids [aa] 1 to 48) in measles virus is the key region for the formation of the $N^0$-P complex (41), and the N protein interacts with P (aa 1 to 50) in PPRV (see Fig. S1 in the supplemental material). Therefore, it was hypothesized that ISG15 could inhibit the formation of the $N^0$-P complex. The

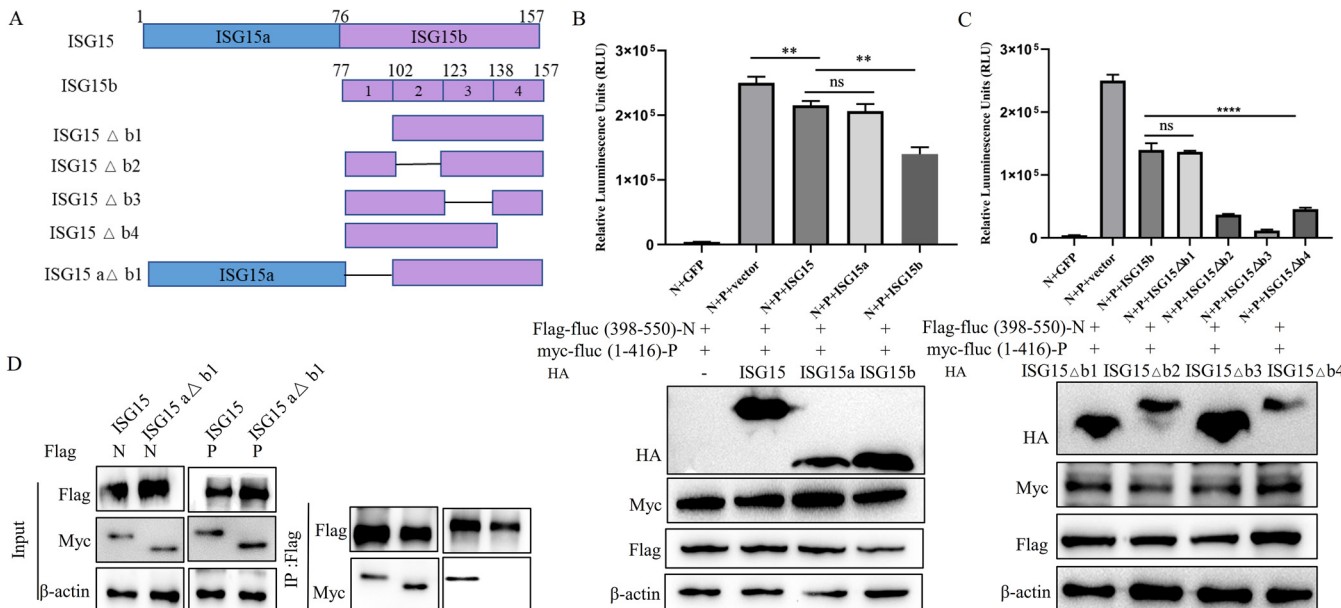

**FIG 8** The aa 77 to 101 of ISG15 interfere with the interaction between N and P proteins. (A) Schematic representation of the strategy for constructing truncated ISG15. (B) According to the luciferase complementary system, HEK-293T cells were cotransfected with Flag-fluc (aa 398 to 550)-N, Myc-fluc (aa 1 to 416)-P, and HA empty or HA-ISG15 or truncated HA-ISG15 aa 1 to 76 and 77 to 157. At 24 hpt, cell lysates were collected to detect luciferase intensity and detected using Western blotting with anti-Flag, anti-Myc, anti-HA, and anti-$\beta$-actin. (C) According to the luciferase complementary system, HEK-293T cells were cotransfected with Flag-fluc (aa 398 to 550)-N, Myc-fluc (aa 1 to 416)-P, and HA empty or truncated HA-ISG15 aa 77 to 157, 102 to 157, 77 to 102, and 123 to 157; 77 to 123 and 138 to 157; and 77 to 138. At 24 hpt, cell lysates were collected to detect luciferase intensity and detected using Western blotting with anti-Flag, anti-Myc, anti-HA, and anti-$\beta$-actin antibodies. (D) HEK-293T cells were cotransfected with Flag-N and Myc-ISG15 or truncated Myc-ISG15 (aa 102 to 157) or Flag-P and Myc-ISG15 or truncated Myc-ISG15 (aa 1 to 76 and 102 to 157). At 24 hpt, cells were harvested, immunoprecipitated with anti-Flag antibody, and further detected using immunoblot analysis with anti-Myc or anti-Flag antibodies. Expression levels of the proteins were analyzed by immunoblot analysis of the lysates with anti-Flag, anti-Myc, and anti-$\beta$-actin antibodies. Data were presented as means from three independent experiments and were statistically evaluated by Student's $t$ test. **, $P < 0.01$; ****, $P < 0.0001$.

luciferase results showed that ISG15 weakened the interaction between N protein and P (aa 1 to 50) (Fig. 7G). Collectively, our results showed that ISG15 can competitively bind P protein with N protein, thus affecting the formation of the $N^0$-P complex.

**ISG15 (aa 77 to 101) interferes with the interaction between N and P proteins.** ISG15 has two ubiquitin-like structural domains, an N-terminal ubiquitin-like domain and a C-terminal ubiquitin-like domain. To identify which domain of ISG15 might be responsible for the inhibition of N and P protein interactions, we generated two mutants (Fig. 8A). HEK-293T cells were cotransfected with the two mutants and the luciferase complementary system plasmid. The luciferase results showed that C-terminal ubiquitin-like domain exhibited reduced luciferase activity, suggesting that this domain plays an important role in the inhibition of interactions between N and P proteins (Fig. 8B). Then, four deletion mutants with the C-terminal ubiquitin-like domain were constructed (Fig. 8A). It was found that there was no significant difference in the inhibition of interactions between N and P proteins in the C-terminal domain mutants with the deletion of amino acids 77 to 101 (Fig. 8C). These results suggested that ISG15 (aa 77 to 101) plays a determining role in the inhibition of interactions between N and P proteins by ISG15. Immunoprecipitation results showed that ISG15, with the deletion of amino acids 77 to 101, still interacts with the N protein but not with the P protein (Fig. 8D). These results suggest that ISG15 (aa 77 to 101) is essential for their binding to the P protein.

## DISCUSSION

It is well-known that PPRV infections are associated with immunosuppression in natural hosts (42). However, a live attenuated vaccine, named Nigeria/75/1, against PPRV is widely used in small ruminants and demonstrates good performance (43). At present, research on the innate immune response to PPRV is not yet comprehensive. PPRV can proliferate in

both Vero and EEC cells, but the virus titers in EEC cells are significantly lower than those in Vero cells (44). This is because Vero cells are IFN deficient and IFN-$\beta$ is significantly upregulated after PPRV infection in EECs. It was suspected that PPRV replication is significantly affected by IFN-stimulating genes (ISGs), which have a wide range of antiviral effects. Previous studies found that among several common ISGs (GBP2, IFIT1, IFIT5, ISG20, MX1, and ISG15), only ISG15 was significantly upregulated after PPRV infected EEC cells (45). ISG15 is considered to be a potent host effector molecule with resistance to a wide range of viruses. Concerning other paramyxoviruses, ISG15 is upregulated by RSV and inhibits RSV proliferation through ISGylation (28). Canine distemper virus infection results in the increased expression of antiviral effector proteins such as MX, PKR, OAS1, and ISG15, which may be associated with restricted viral replication (46). The overexpression of ISG15 has an antiviral effect on caprine parainfluenza virus type 3 (47). We demonstrated that ISG15 inhibited the proliferation of PPRV in EECs. However, it did not prevent the proliferation of PPRV in EECs, indicating that some mechanism exists that blocks the inhibitory effect of ISG15.

ISG15 can bind to host and viral proteins in conjugated form, and free intracellular ISG15 can function as a cytokine and noncovalently bind to intracellular proteins and modulate their functions (48). LRLRGG is required for ISGylation at the C terminus of ISG15. In this study, after the ISG15 C-terminal GG was mutated to AA, the replication of PPRV was still inhibited, but the effect was weaker than that of the nonmutated group. These results suggest that ISG15 inhibits PPRV replication without relying solely on its ISGylation. The same conclusion was reached in Vero-SLAM cells. Both free ISG15 and ISGylation may play an inhibitory role in the proliferation of PPRV. It has been shown that the expression of free ISG15 or the ISGylation system (UbE1L and UbcH8) inhibits the budding of the Ebola virus VP40 virus-like particles (VLPs) (37). Previous studies have shown that both ISGylation and free ISG15 can have antiviral effects, with the mechanisms of impact differing among different viruses.

In the current study, we first excluded free ISG15 that did not participate in the virus binding and entry. Next, we demonstrated through a PPRV minigenome assay that ISG15 can interfere with PPRV RNP complex activity. In addition, we revealed that ISG15 specifically inhibited viral RNA replication and did not affect viral mRNA transcription. The viral RNA genome and antigenome of *Morbilivirus* are coated by the viral N protein, and the competent replication of viral genomic and complementary genomic RNA depends on the N protein, which must be maintained as monomeric and RNA-free (N$^0$) to specifically encapsidate these replication products as they are synthesized (8, 49). Studies have shown that the C terminus of the P protein binds N-RNA and that the N terminus of soyuz1 binds to the N core domain to form the N$^0$-P complex (50), which performs a chaperone function to prevent newly synthesized N proteins from binding to cellular RNA and regulates the viral transcription and replication process (9, 10). In a previous report, ISG15 was shown to disrupt the viral replication machinery by binding to influenza B virus nucleoproteins, disrupting the ability of the virus to oligomerize and form viral nucleoproteins, thereby inhibiting viral RNA synthesis (24).

In this study, we demonstrated that free ISG15 can interact with the N termini of both N and P proteins and can attenuate the interaction of N and P proteins by competitively binding to them, as ISG15 binds to P (aa 1 to 50), which is the region that can bind N$^0$, thus disrupting the formation of the N$^0$-P complex and the balance between transcription and replication and affecting the replication of PPRV. It was found that peptides designed according to the P protein of RSV could inhibit viral replication *in vitro* and *in vivo* by preventing the formation of the N$^0$-P complex (51). The interaction of the N protein with the C terminus of the P protein was not detected in this study, which may be due to the low affinity of N protein interactivity with the P protein to form replication complexes. In the measles virus, it has been shown that the formation of the replication complex occurs via the binding of N (aa 477 to 505) to P (aa 457 to 507) proteins in a low-affinity interaction (41, 50). For the mumps virus, a binding site for the P protein (aa 343 to 391) maps to the assembly domain of the N protein (aa 1 to 398), while no strong binding of the P protein

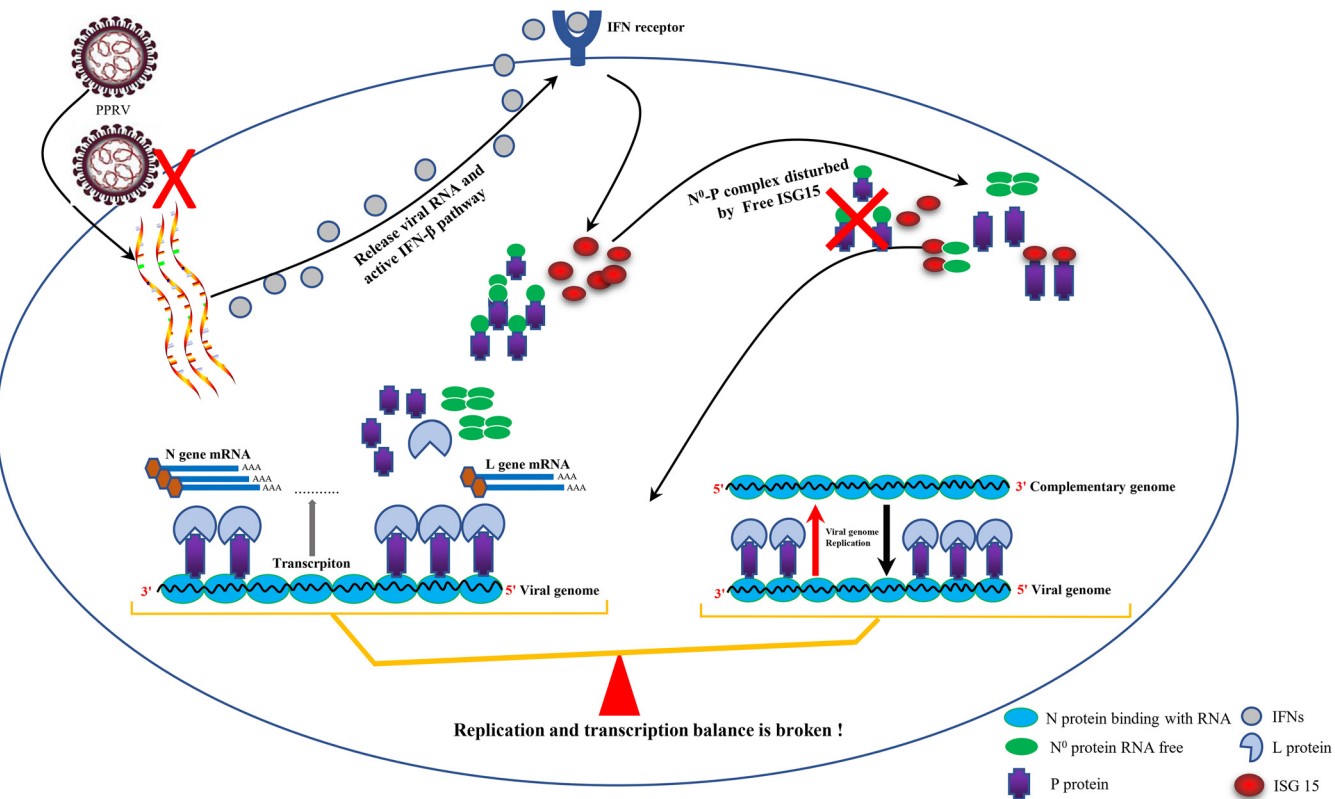

**FIG 9** Proposed working model that illustrates that ISG15 inhibits PPRV replication. During PPRV infection, ISG15 binds competitively to the N and P protein N termini, disrupting the formation of the $N^0$-P complex and disrupting the balance of viral transcription and replication, thereby affecting PPRV replication.

to the tail of the N protein was detected (50). These results suggest that there may be some significant differences among segments of the negative-strand RNA virus.

By constructing ISG15 mutants of different lengths, we further identified ISG15 (aa 77 to 101) that plays a critical role in blocking the formation of the $N^0$-P complex, which is located at the hinge area between two UBL domains of ISG15 (52). Interestingly, according to the three-dimensional model of ISG15 (Fig. 9), aa 77 to 101 is a knot-like structure that closely resembles a well-known antiviral peptide cyclotide (53). As that aa 77 to 101 have a more obvious effective function than full-length ISG15, it may be that short ISG15 much more easily exposes the active domain of aa 77 to 101 to block the formation of the $N^0$-P complex.

In summary, we found that free ISG15 inhibits PPRV replication by attenuating the interaction between N and P proteins. These findings provide a new perspective that can further the understanding of the interference of ISG15 in the binding of N and P proteins, which could represent an effective anti-PPRV strategy, one that can yield important insights into the pathogenesis and immune mechanisms of PPRV.

## MATERIALS AND METHODS

**Cells and virus.** Caprine endometrial epithelial cells (EECs), which were kindly provided by Yaping Jin (Northwest Agriculture and Forestry University, Yangling, Shaanxi, China), were cultured in Dulbecco's minimal essential medium-Ham's F-12 nutrient mixture medium (DMEM-F12; Gibco) with 10% fetal bovine serum (FBS; Gibco). Human embryonic kidney (HEK-293T), BSR, and Vero-SLAM cells were cultured in DMEM with 10% FBS. All cells were cultured at 37°C with 5% $CO_2$. The vaccine strain PPRV Nigeria/75/1 was propagated in Vero-SLAM cells.

**qRT-PCR.** Analysis by reverse transcription-quantitative PCR (qRT-PCR) was performed to detect the expression of ISG15, IFN-$\beta$, and PPRV mRNA. All primers are listed in Table 1. Total RNA was extracted from EECs using TRIzol (Invitrogen) and transcribed into cDNA according to the Novozymes PrimeScript RT reagent kit (Vazyme). To detect the three viral RNA species produced by PPRV, reverse transcription was conducted using oligo(dT) primers and N gene-specific oligonucleotides (RT-vRNA and RT-cRNA) for

**TABLE 1** Primers used for plasmid constructs, reverse transcription, and qRT-PCR

| Primer name | Sequence (5′–3′) |
|---|---|
| GAPDH-qF(goat) | TGGAGAAACCTGCCAAGTATG |
| GAPDH-qR(goat) | TGAGTGTCGCTGTTGAAGTC |
| ISG15-qF | CAATGTGCCTGCTTTCCAG |
| ISG15-qR | ACCCTTGTCGTTCCTCACC |
| IFNβ-qF(goat) | TGCCTCCTCCAGATGGTTCTCC |
| IFNβ-qR(goat) | TGCTGTGCTTGCTTCATCTCCTC |
| PPRV-N-qF | GTTATCATAGTCCCCATTCCCG |
| PPRV-N-qR | CTCCACGAACAAAGATAACATGC |
| myc-ISG15-F | CCGGAATTCGGGCCACCATGGGCGGGGACCTGAAGGTGAAG |
| myc-ISG15-R | CCG CTCGAG CTACCCACCCCGCAGACGTAG |
| plov-GFP-ISG15-F | gatgatgacgacaaagctagcATGGGCGGGGACCTGAAG |
| plov-GFP-ISG15-R | ccctcgacgcctagggcggccgcCTACCCACCCCGCAGACG |
| ISG15-sgRNA-F | CACC GCCTGGCCCACCTTGACAGC |
| ISG15-sgRNA-R | AAAC GCTGTCAAGGTGGGCCAGGC |
| ISG15(AA)-F | aatctacgtctgcgggCtgCgtagCTCGAGG |
| ISG15(AA)-R | CGCAGCCCGCAGACGTAGATTCATGAACACG |
| 3xflag-NF (1–400 aa) | caagcttgcggccgcgaattcA ATGGCTACTCTCCTTAAAAGCTTGG |
| 3xflag-NR (1–400 aa) | cctctagagtcgactggtaccTCACTGTGAGGCGATTTCCGAGA |
| 3xflag-NF (401–525 aa) | caagcttgcggccgcgaattcA ATGACTGGGGATGAACGAACCG |
| 3xflag-NR (401–525 aa) | cctctagagtcgactggtaccTCAGCTGAGGAGATCCTTGTCG |
| myc-PF (1–377 aa) | tggccatggaggcccgaattcGG ATGGCAGAAGAACAAGCATACCA |
| myc-PR (1–377 aa) | ccgcggccgcggtacctcgag TTAAAAGCCCGGGATGGCTATC |
| myc-PF (378–509 aa) | tggccatggaggcccgaattcGG ATGGGGAAGGACATCAAGGACCC |
| myc-PR (378–509 aa) | ccgcggccgcggtacctcgag TTACGGCTGCTTGGCAAGA |
| GST-ISG15-F | gatctggttccgcgtggatccATGGGCGGGGACCTGAAG |
| GST-ISG15-R | ctcgagtcgacccgggaattcCTACCCACCCCGCAGACG |
| GST-N-F | gatctggttccgcgtggatccATGGCTACTCTCCTTAAAAGCTTGG |
| GST-N-R | ctcgagtcgacccgggaattcTCAGCTGAGGAGATCCTTGTCG |
| GST-P-F | gatctggttccgcgtggatccATGGCAGAAGAACAAGCATACCA |
| GST-P-R | ctcgagtcgacccgggaattcTTACGGCTGCTTGGCAAGA |
| vRNA-R (671–696 bp) | aaccgccgaagtgatagatcttctg |
| cRNA-R (3–27 bp) | ggctactctccttaaaagcttggca |

mRNA, vRNA, and cRNA, respectively, by using Moloney murine leukemia virus (M-MLV) reverse transcriptase kits (Promega, USA) (Table 1). The qRT-PCR was performed using Novozymes Ultra SYBR mixture (Vazyme). Gene expression was normalized to the GAPDH (glyceraldehyde-3-phosphate dehydrogenase) expression, and the comparative cycle threshold ($\Delta\Delta C_T$) method was used for relative quantifications.

**Western blotting.** Western blotting was used to analyze protein expression. Protein concentrations were determined using the Beyotime protein quantification kit, and samples of the same total protein amounts were separated by 12% SDS-PAGE and subsequently transferred to polyvinylidene fluoride membranes and blocked by 5% skimmed milk at 4°C overnight. Membranes were immersed in prepared primary antibodies, which included a mouse anti-$\beta$-actin monoclonal antibody (Kangwei), a mouse anti-Flag (Sigma), a mouse anti-Myc (Abcam), a rabbit anti-ISG15, and a mouse anti-N (prepared in our laboratory). Primary antibodies were incubated for 2 h at room temperature, after which horseradish peroxidase-conjugated anti-rabbit or anti-mouse Ig (Sigma) was used as a secondary antibody. Protein bands were incubated with enhanced chemiluminescence (Invitrogen) and analyzed using an image analysis system (Tanon).

**Immunofluorescence assay.** To analyze the colocalization of ISG15 with PPRV proteins, EECs after cotransfection with plasmids or infection with PPRV were fixed with 4% paraformaldehyde for 30 min at room temperature and washed three times with PBS for 6 min each. The cells were permeabilized by prechilled methanol at −20°C and incubated at the same temperature for 10 min. The cells were blocked with 5% bovine serum albumin and 0.3% Triton X-100 at room temperature for 2 h. Primary antibodies and secondary antibodies (Alexa Fluor 594 goat anti-rabbit immunoglobulin G [H+L] and Alexa Fluor 488 goat anti-mouse immunoglobulin G [H+L]; Invitrogen) were then added. Nuclei were stained with 4′,6-diamidino-2-phenylindole (DAPI; Invitrogen). Images were taken with a Zeiss LSM880 confocal microscope (Zeiss).

**ISG15 overexpression assays.** Total RNA was extracted from EECs, reverse transcribed with random primers, and amplified with the primers listed in Table 1, after which the PCR products were ligated to a Plov-CMV-GFP vector by the infusion method, according to the instructions of the ClonExpress II one-step cloning kit (Vazyme), and sequenced to verify whether the ISG15 sequence was correct. The correctly sequenced plasmid was cotransfected with the lentiviral packaging plasmids psPAX2 and pMD2.G into HEK-293T cells at a density of ∼80%, according to the Lipofectamine 3000 instructions (Invitrogen). DMEM containing 10% FBS was replaced 6 h after transfection. Cell supernatants were collected at 24 h and 48 h posttransfection. The supernatant containing the lentivirus was infected with EECs in 6-well plates, positive cells were screened with a final concentration of 2 $\mu$g/mL of puromycin (Invitrogen), and

the green fluorescence in the positive cells was observed with a fluorescence microscope and analyzed using Western blotting with an N-terminal Flag tag and ISG15 antibody.

**siRNA silencing and knockout of the ISG15 assays.** Twenty-four hours before transfection, EECs were placed in 24-well plates with $10^5$ cells per well and transfected with SIG15 siRNA and control siRNA. For subsequent experiments, siRNA 1 was selected with a final concentration of 100 nM. At 24 h after transfection, cells were infected with MOI of 1 of PPRV. At 48 h after infection, samples were harvested for qRT-PCR, Western blotting, and virus titration assay.

Plasmids containing sgISG15 were cotransfected into HEK-293T cells by Lipofectamine 3000 along with ancillary plasmids (psPAX2 and pMD2.G). At 6 h after transfection, the medium was changed to a complete medium containing 10% FBS. The supernatant was collected at 24 and 48 h posttransfection. Lentivirus was added to EECs in 6-well culture plates. The cells were then incubated for 48 h. Puromycin was used to screen positive cells in a complete growth medium at a final concentration of 2 $\mu$g/mL. Knockout of ISG15 was confirmed by Western blotting.

**Cell viability assays.** Cell viability assays were performed with the Cell Counting Kit-8 (CCK-8) (Beyotime) according to the manufacturer's instructions.

**Virus titration assay.** Vero-SLAM cells were inoculated with the collected virus samples in 96-well plates with 10 groups generated by a 10-fold dilution series ($10^{-1}$ to $10^{-10}$); one longitudinal row of eight wells was inoculated at each dilution, 100 $\mu$L per well. Cells not inoculated with PPRV were used as a negative control. The virus solution was discarded 2 h after incubation, and 200 $\mu$L of medium containing 2% FBS was added and incubated at 37°C, with the results observed and recorded daily for 7 days and calculated according to the Reed-Muench method.

**PPRV binding and entry.** EECs ($n = 10^5$) were spread into 12-well plates and cultured for 24 h, after which they were infected with PPRV (MOI of 5) and incubated for 1 h at 4°C. For virus binding assays, cells were washed twice with precooled PBS, pH 1.3, to remove unbound virus and were harvested for detection of viral RNA and proteins by qRT-PCR and Western blotting, respectively. For entry assays, after incubation at 4°C for 1 h to allow for virus binding, cells were washed twice with precooled PBS, pH 1.3, and then incubated with prewarmed DMEM at 37°C for 10 min. Cells were washed three times with PBS, pH 1.3, to remove the attached but not yet internalized virus. Total cellular RNA was extracted, and viral RNA and proteins were quantified by qRT-PCR and Western blotting, respectively.

**Flow cytometry.** BSR cells were transfected with a viral minigenome (full length of viral N, P, and L proteins and plasmids with reverse insertion of GFP) and ISG15 or an empty plasmid. After 36 h, green fluorescence was observed under fluorescence microscopy, and the cells were digested with trypsin and washed three times with PBS. Cells with green fluorescence were screened by flow cytometry.

**Coimmunoprecipitation.** After cotransfection for 36 h, the protein supernatant was collected by centrifugation at 4°C at 12 000 $g$ after the cells were lysed with the cell lysis solution in the coimmunoprecipitation (co-IP) kit (Invitrogen). Finally, analysis was performed by Western blotting.

**Prokaryotic expression and purification of recombinant proteins.** All glutathione *S*-transferase (GST)-tagged proteins were expressed in *Escherichia coli* BL21(DE3)-competent cells (TransGen). A bacterial solution (200 mL) was incubated at 37°C until the mid-log phase (optical density at 600 nm [OD$_{600}$], 0.6 to 0.8). The final concentration of 1 mM isopropyl-$\beta$-D-thiogalactopyranoside was added and incubated at 16°C for 24 h. The bacterial precipitate was collected by centrifugation at 6,000 $\times$ $g$, and proteins were purified using a GST tag protein purification kit (Beyotime Biotechnology).

**GST pulldown assays.** *In vitro* binding assays were performed in HEK-293T cells expressing Flag-tagged N, P, and their truncated proteins according to the manufacturer's instructions (GST protein interaction pulldown kit; Invitrogen) and analyzed by Western blotting.

**Competition assays.** The HEK-293T cells were placed in 12-well plates in triplicate 24 h before transfection. Five hundred nanograms of Flag-fluc (aa 398 to 550)-N (the C-terminal fragment [aa 398 to 550] of firefly luciferase [Luc] was cloned into Flag-N) and Myc-fluc (1 to 416)-P (the N-terminal fragment [aa 1 to 416] of Luc was cloned into myc-P) were transfected with 1,000 nanograms of HA empty or HA-ISG15. Then, 24 h later, cell lysates were collected by passive lysis buffer (Promega). A luciferase assay was performed according to instructions (Promega).

**Statistical analysis.** Statistical significance was determined with the Student $t$ test ($P < 0.05$) between two groups. One-way analysis of variance followed by Tukey's multiple-comparison test and two-way analysis of variance followed by Sidak's multiple-comparison test were used to compare multiple groups in GraphPad Prism 8 ($>$2).

## SUPPLEMENTAL MATERIAL

Supplemental material is available online only.

**SUPPLEMENTAL FILE 1**, PDF file, 0.2 MB.

## ACKNOWLEDGMENTS

This study was supported by the National Natural Science Foundation of China (no. 32172832, no. 32000109), Shanghai Sailing Program (20YF1457700), the China Postdoctoral Science Foundation (no. 2019M660885, no. 2021T140718), and the Central Public-interest Scientific Institution Basal Research Fund (2021JB08).

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
