## [Reviewer comments · Microbiology Spectrum]

Microbiology Spectrum

Free ISG15 Inhibits the Replication of Peste des petits Ruminants Virus by Breaking the Interaction of Nucleoprotein and Phosphoprotein

Guangqing liu, Jingyu Tang, Aoxing Tang, Nannan Jia, Hanyu Du, Chuncao Liu, Jie Zhu, Chuanfeng Li, and Chunchun Meng

Corresponding Author(s): Guangqing liu, Shanghai Veterinary Research Institute

Review Timeline:

Submission Date:	March 26, 2022
Editorial Decision:	May 24, 2022
Revision Received:	July 23, 2022
Accepted:	August 12, 2022

Editor: Daniela Rajao

Reviewer(s): Disclosure of reviewer identity is with reference to reviewer comments included in decision letter(s). The following individuals involved in review of your submission have agreed to reveal their identity: Haixue Zheng (Reviewer #2); David John Hughes (Reviewer #3)

Transaction Report:

DOI: <https://doi.org/10.1128/spectrum.01031-22>

May 20, 2022

Prof. Guangqing liu
Shanghai Veterinary Research Institute
Chinese Academy of Agricultural Sciences
No.518 Ziyue Road
shanghai
China

Re: Spectrum01031-22 (Free ISG15 Inhibits the Replication of Peste des petits Ruminants Virus by Breaking the Interaction of Nucleoprotein and Phosphoprotein)

Dear Prof. Guangqing liu:

Link Not Available

Sincerely,

Daniela Rajao

Journals Department
Reviewer comments:

Reviewer #1 (Comments for the Author):

The authors investigated the role of ISG15 in PPRV infection and replication. They found that ISG15 could negatively regulate replication of PPRV by directly disrupting interaction between free N and P. Overall, the work was technically well designed. However, the paper is generally poorly written. It should be modified by an English editing service to improve the readability.

1) Here are some examples of vague expression.

- From line 221 to 223, the results in three figures were described in one sentence using Fig 4D and E (should be 4E?) as subject. It is confusing.
 - From line 232 to 238, the description of the virus binding and entry assay is confusing. And it seems there is no need to repeat the description both in Materials and Methods and Results.
 - In line 248, "The level of fluorescence in transfected ISG15 cells". Does it mean "The level of fluorescence in ISG15-transfected cells"?
 - In line 204, "PPRV titers in culture supernatants of EEC-GFP were significantly decreased compared with those in EEC-GFP at 24 and 48 hpi (Fig.3F)". Should the first EEC-GFP be EEC-ISG15?
 - From line 289 to 293, the extremely long sentence should be rewritten to make the meaning clear.
- 2) In Discussion, the results were redescribed by quoting the figures (line 330, 349, 360 and 376). I suggested the authors focus on discussion of the results in this part.
- 3) Nigeria 75/1 is more widely used for the vaccine strain. Please unify the name of the vaccine strain. Both Nigeria/75/1 (in line 411) and N75 (in line 319) were used by the authors.
- 4) In line 320, it is not precisely to say "As most PPRV isolates cannot be cultured in vitro". Lots of PPRV isolates have been successfully in vitro cultured in different laboratories worldwide.

Reviewer #2 (Comments for the Author):

The manuscript entitled "Free ISG15 Inhibits the Replication of Peste des petits Ruminants Virus by Breaking the Interaction of Nucleoprotein and Phosphoprotein " suggested that ISG15 expression was significantly upregulated in caprine endometrial epithelial cells (EECs) after PPRV infection. Knockdown or knockout of ISG15, viral proliferation was significantly promoted. ISG15 inhibited PPRV replication independent of its ISGylation activity, and detailed analysis revealed that ISG15 interacted and co-localized with both viral N and P proteins. Its interactive regions were all located in the N-terminal domain. Luciferase complementation experiment confirmed that ISG15 can competitively interact with N and P proteins and significantly interfere with their binding. Finally, the authors found that the 77-101 amino acids (aa) region of ISG15 played a key role in inhibiting the binding of N and P proteins, and the interaction of ISG15 with the P protein disappeared after the deletion of 77-101aa. The experiment design is logical and the results is interesting.

Major comment:

In fig 6 the author only identified that the 1-377 aa region of P interacted with ISG15 and concluded that ISG15 interfered with the N0-P formation. In previous publication , there are only the N-terminal of P (about 1-50 or 70 aa) responsible for the N0-P complex (Targeting the Respiratory Syncytial Virus N0-P Complex with Constrained α -Helical Peptides in Cells and Mice;Structure of the Vesicular Stomatitis Virus N0-P Complex) formation in almost all of the non-segment negative RNA virus, therefore, the detailed region in P protein of PPRV responsible for P-ISG15 interaction should be identified. Meanwhile, the interaction of endogenous ISG15 with PPRV N and P should be confirmed.

Minor comments:

1. The manuscript contains many instances of inappropriate language and grammar usage,, for example, line 48: "suppresses", line 186: "were" used wrong tenses, which need to be corrected.
2. Consider replacing proliferation with replication in the article.
3. Line 349: It would be better to write, Both free ISG15 and ISGylation may play.....
4. It is recommended to write all "myc" in the article and pictures as "Myc".
5. Both "co-transfected" and "cotransfected" are used in the article, and one expression is recommended.
6. In the article, abbreviations should be placed where they are first mentioned, such as amino acids(aa).
7. The labels of figures should be check carefully and in a unified form.
8. In the hypothesis model, the interaction between ISG and PPRV N should also be revealed in the image.

Reviewer #3 (Comments for the Author):

PPRV is an extremely important morbillivirus that causes high levels of morbidity and mortality in infected animals. The work presented in this manuscript describes a novel function for an antiviral restriction factor, ISG15, that is expressed upon infection. Interestingly, the authors demonstrate that ISG15, a ubiquitin-like protein that can have antiviral activity through covalent

modification, restricts PPRV infection through non-covalent modifications. This is interesting because ISGylated proteins are usually at significantly lower concentrations than their unmodified counterparts, which often makes interpretation of ISG15's antiviral activity challenging. This potentially new mechanism could be at play for a number of viruses. The degree of antiviral activity of free ISG15, particularly when it came to viral titres, differences were usually trivial, making it difficult to be enthusiastic about this manuscript. There are additional issues that I feel should be addressed before it is suitable for publication, if accepted:

1. I am concerned that most data are not derived from biological replicates. Information on how many biological replicates were used to calculate statistical data needs to be stated in figure legends. What method was used to calculate error? What statistical test was used? How many times were qualitative data performed? A particularly confusing issue was that statistical differences in Fig. 2E were calculated using t-test and 2-way ANOVA. Which was it and if ANOVA, what post test was used to determine differences between groups?
2. Differences in PPRV titres following siRNA knockdown of ISG15 are negligible (seemingly only 1-1.5-fold different). Can the authors be confident the siRNA is functional, particularly as it wasn't validated in either IFN-treated or infected cells? It is known (and shown on Fig. 1F of the manuscript) that ISG15 expression in unstimulated cells is barely detectable.
3. How were the data in Fig. 2D analysed? Figure legends need to provide necessary detail. Presumably, levels are compared to 'No treat' which is set to 1. If so, it is troublesome that there has been a blotting transfer issue with that sample. Again, differences are small, and it is not known if this difference is statistically different.
4. Fig. 2D: ISG15 knockout data are not convincing. There is a band, although poorly transferred, at the same MW as ISG15. Previous data have shown the ISG15 antibody to be specific, so new data should be provided (particularly as in the text it is stated that ISG15 is 'undetected' in these cells [line 185]).
5. Overexpression of ISG15 led to minimal differences in viral titre compared to controls ($<1 \text{ Log}_{10} \text{ TCID}_{50}$). It's not clear how many biological replicates these data were derived from. Would this difference hold when comparing biological replicates?
6. Data in Fig. 5G need annotating. As they stand, they don't mean anything. Again, differences are negligible. Are the data from biological replicates?
7. Immunofluorescence data in Fig. 6E shows many, if not the majority, of the cells express nuclear N and P. This is unusual.
8. ISG15-dependent inhibition of N-P protein interactions differ significantly between the two assays in Fig. 7E & F. In particular, the luciferase assay produced negligible differences. Does this explain the small differences observed in viral titres?

Minor:

The manuscript should be written in past tense.
Line 235-238 - text is confusing.

Staff Comments:

Preparing Revision Guidelines

Please return the manuscript within 60 days; if you cannot complete the modification within this time period, please contact me. If you do not wish to modify the manuscript and prefer to submit it to another journal, please notify me of your decision

immediately so that the manuscript may be formally withdrawn from consideration by Microbiology Spectrum.

Review on Spectrum01031-22

The authors investigated the role of ISG15 in PPRV infection and replication. They found that ISG15 could negatively regulate replication of PRRV by directly disrupting interaction between free N and P. Overall, the work was technically well designed. However, the paper is generally poorly written. It should be modified by an English editing service to improve the readability.

- 1) Here are some examples of vague expression.
 - From line 221 to 223, the results in three figures were described in one sentence using Fig 4D and E (should be 4E?) as subject. It is confusing.
 - From line 232 to 238, the description of the virus binding and entry assay is confusing. And it seems there is no need to repeat the description both in Materials and Methods and Results.
 - In line 248, “The level of fluorescence in transfected ISG15 cells”. Does it mean “The level of fluorescence in ISG15-transfected cells”?
 - In line 204, “PPRV titers in culture supernatants of EEC-GFP were significantly decreased compared with those in EEC-GFP at 24 and 48 hpi (Fig.3F)”. Should the first EEC-GFP be EEC-ISG15?
 - From line 289 to 293, the extremely long sentence should be rewritten to make the meaning clear.
- 2) In Discussion, the results were redescribed by quoting the figures (line 330, 349, 360 and 376). I suggested the authors focus on discussion of the results in this part.
- 3) Nigeria 75/1 is more widely used for the vaccine strain. Please unify the name of the vaccine strain. Both Nigeria/75/1 (in line 411) and N75 (in line 319) were used by the authors.
- 4) In line 320, it is not precisely to say “As most PPRV isolates cannot be cultured *in vitro*”. To my knowledge, lots of PPRV isolates have been successfully *in vitro* cultured in different laboratories worldwide.

Dear respected editor :

Thank you very much for inviting reviewers to review our article and giving valuable comments. We admire your expertise and patient. The comments are very useful for our research. We decide to accept all the comments and revise our manuscript carefully according to all comments. We look forward to working with you and the reviewers to move this manuscript closer to publication in Microbiology Spectrum.

The following is our response to reviewers.

Reviewer comments:

Reviewer #1 (Comments for the Author):

The authors investigated the role of ISG15 in PPRV infection and replication. They found that ISG15 could negatively regulate replication of PPRV by directly disrupting interaction between free N and P. Overall, the work was technically well designed. However, the paper is generally poorly written. It should be modified by an English editing service to improve the readability.

Reply: Thanks for the comments. We already polished our manuscript with an English editing service.

1) Here are some examples of vague expression.

- From line 221 to 223, the results in three figures were described in one sentence using Fig 4D and E (should be 4E?) as subject. It is confusing.

Reply: Thanks for the comments. This issue has been modified in lines 211-214.

- From line 232 to 238, the description of the virus binding and entry assay is confusing. And it seems there is no need to repeat the description both in Materials and Methods and Results.

Reply: Thanks for the comments. The relevant description has been removed from the results.

- In line 248, "The level of fluorescence in transfected ISG15 cells". Does it mean

"The level of fluorescence in ISG15-transfected cells"?

Reply: Thanks for the comments. This issue has been modified in line 236.

- In line 204, "PPRV titers in culture supernatants of EEC-GFP were significantly decreased compared with those in EEC-GFP at 24 and 48 hpi (Fig.3F)". Should the first EEC-GFP be EEC-ISG15?

Reply: Thanks for the comments. The first EEC-GFP has been modified to EEC-ISG15 in line 194.

- From line 289 to 293, the extremely long sentence should be rewritten to make the meaning clear.

Reply: Thanks for the comments. This issue has been modified in lines 276-280.

2) In Discussion, the results were redescribed by quoting the figures (line 330, 349, 360 and 376). I suggested the authors focus on discussion of the results in this part.

Reply: Thanks for the comments. This issue has been modified in lines 317-319, 335, 345, and 359.

3) Nigeria 75/1 is more widely used for the vaccine strain. Please unify the name of the vaccine strain. Both Nigeria/75/1 (in line 411) and N75 (in line 319) were used by the authors.

Reply: Thanks for the comments. The vaccine strains in the article have been standardized to Nigeria/75/1.

4) In line 320, it is not precisely to say "As most PPRV isolates cannot be cultured in vitro". Lots of PPRV isolates have been successfully in vitro cultured in different laboratories worldwide.

Reply: Thanks for the comments. We have modified our statement in the revised manuscript.

Reviewer #2 (Comments for the Author):

The manuscript entitled "Free ISG15 Inhibits the Replication of Peste des petits Ruminants Virus by Breaking the Interaction of Nucleoprotein and Phosphoprotein " suggested that ISG15 expression was significantly upregulated in caprine endometrial epithelial cells (EECs) after PPRV infection. Knockdown or knockout of ISG15, viral proliferation was significantly promoted. ISG15 inhibited PPRV replication independent of its ISGylation activity, and detailed analysis revealed that ISG15 interacted and co-localized with both viral N and P proteins. Its interactive regions were all located in the N-terminal domain. Luciferase complementation experiment confirmed that ISG15 can competitively interact with N and P proteins and significantly interfere with their binding. Finally, the authors found that the 77-101 amino acids (aa) region of ISG15 played a key role in inhibiting the binding of N and P proteins, and the interaction of ISG15 with the P protein disappeared after the deletion of 77-101aa. The experiment design is logical and the results is interesting.

Major comment:

In fig 6 the author only identified that the 1-377 aa region of P interacted with ISG15 and concluded that ISG15 interfered with the N0-P formation. In previous publication, there are only the N-terminal of P (about 1-50 or 70 aa) responsible for the N0-P complex (Targeting the Respiratory Syncytial Virus N0-P Complex with Constrained α -Helical Peptides in Cells and Mice; Structure of the Vesicular Stomatitis Virus N0-P Complex) formation in almost all of the non-segment negative RNA virus, therefore, the detailed region in P protein of PPRV responsible for P-ISG15 interaction should be identified. Meanwhile, the interaction of endogenous ISG15 with PPRV N and P should be confirmed.

Reply: Thanks for the comments. We already through continuous truncation of P protein, it was found that N protein interacted with P protein aa 1 to 50(Fig. S1), and ISG15 affected the interaction between N protein and P aa 1 to 50(Fig. 7G). The details are described in lines 283-288. Meanwhile, the Co-IP assay found that ISG15

can interact with endogenous N and P proteins (Fig. 6G and 6H), which was described in lines 258-260.

Minor comments:

1. The manuscript contains many instances of inappropriate language and grammar usage, for example, line 48: "suppresses", line 186: "were" used wrong tenses, which need to be corrected.

Reply: Thanks for the comments. These issues have been modified in the article.

2. Consider replacing proliferation with replication in the article.

Reply: Thanks for the comments. The part proliferation in the article has been modified to replication.

3. Line 349: It would be better to write, Both free ISG15 and ISGylation may play.....

Reply: Thanks for the comments. This issue has been modified in lines 335-336.

4. It is recommended to write all "myc" in the article and pictures as "Myc".

Reply: Thanks for the comments. All of them have been modified to Myc in the article.

5. Both "co-transfected" and "cotransfected" are used in the article, and one expression is recommended.

Reply: Thanks for the comments. All references to co-transfected have been changed to cotransfected in the article.

6. In the article, abbreviations should be placed where they are first mentioned, such as amino acids(aa).

Reply: Thanks for the comments. The abbreviations in the article have been carefully revised.

7. The labels of figures should be check carefully and in a unified form.

Reply: Thanks for the comments. The labels of the figures have been checked carefully and in a unified form.

8. In the hypothesis model, the interaction between ISG and PPRV N should also be revealed in the image.

Reply: Thanks for the comments. The model diagram has been modified and the interaction between ISG15 and N protein is revealed in Fig. 9.

Reviewer #3 (Comments for the Author):

PPRV is an extremely important morbillivirus that causes high levels of morbidity and mortality in infected animals. The work presented in this manuscript describes a novel function for an antiviral restriction factor, ISG15, that is expressed upon infection. Interestingly, the authors demonstrate that ISG15, a ubiquitin-like protein that can have antiviral activity through covalent modification, restricts PPRV infection through non-covalent modifications. This is interesting because ISGylated proteins are usually at significantly lower concentrations than their unmodified counterparts, which often makes interpretation of ISG15's antiviral activity challenging. This potentially new mechanism could be at play for a number of viruses.

The degree of antiviral activity of free ISG15, particularly when it came to viral titres, differences were usually trivial, making it difficult to be enthusiastic about this manuscript. There are additional issues that I feel should be addressed before it is suitable for publication, if accepted:

1. I am concerned that most data are not derived from biological replicates. Information on how many biological replicates were used to calculate statistical data needs to be stated in figure legends. What method was used to calculate error? What statistical test was used? How many times were qualitative data performed? A particularly confusing issue was that statistical differences in Fig. 2E were calculated using t-test and 2-way ANOVA. Which was it and if ANOVA, what post test was used to determine differences between groups?

Reply: Thanks for the comments. It has been illustrated in the legend that the experimental data were presented as means from three independent experiments. The Fig 2E is calculated by one-way ANOVA, and all the statistical analyses were modified in the revised manuscript.

2. Differences in PPRV titres following siRNA knockdown of ISG15 are negligible (seemingly only 1-1.5-fold different). Can the authors be confident the siRNA is functional, particularly as it wasn't validated in either IFN-treated or infected cells? It

is known (and shown on Fig. 1F of the manuscript) that ISG15 expression in unstimulated cells is barely detectable.

Reply: Thanks for the comments. After ISG15 siRNA interference, both transcript and protein levels of ISG15 were significantly reduced, as demonstrated by quantitative and western blotting assays such as in Fig. 2B and 2D. Therefore, it can be determined that siRNA is functional.

3. How were the data in Fig. 2D analysed? Figure legends need to provide necessary detail. Presumably, levels are compared to 'No treat' which is set to 1. If so, it is troublesome that there has been a blotting transfer issue with that sample. Again, differences are small, and it is not known if this difference is statistically different.

Reply: Thanks for the comments. The ISG15 siRNA group was infected with PPRV after transfection with ISG15 siRNA, the control siRNA group was infected with PPRV after transfection with control siRNA, and the no treat group was infected with PPRV only. According to your suggestion, we experimented again and provided the new figure in Fig. 2D.

4. Fig. 2D: ISG15 knockout data are not convincing. There is a band, although poorly transferred, at the same MW as ISG15. Previous data have shown the ISG15 antibody to be specific, so new data should be provided (particularly as in the text it is stated that ISG15 is 'undetected' in these cells [line 185]).

Reply: Thanks for the comments. The previously provided figure is the EEC-ISG15^{-/-} screened to the 5th generation, and new data have been provided after screening the monoclonal EEC-ISG15^{-/-} (Fig. 2G).

5. Overexpression of ISG15 led to minimal differences in viral titre compared to controls (<1 Log₁₀ TCID₅₀). It's not clear how many biological replicates these data were derived from. Would this difference hold when comparing biological replicates?

Reply: Thanks for the comments. It has been shown in the legend that the experimental data were presented as means from three independent experiments, and this difference hold when comparing biological replicates.

6. Data in Fig. 5G need annotating. As they stand, they don't mean anything. Again,

differences are negligible. Are the data from biological replicates?

Reply: Thanks for the comments. In the minigenome assay, we use N, P, L, PTVT-GFP and ISG15 or vector, a total of 5 plasmids for cotransfection. Only the cells simultaneously containing all the plasmids can express fluorescence. Therefore, the positive percentage is less and causes relatively minor changes. To ensure the accuracy of the experimental results, we performed multiple flow cytometry to compare the differences. The data in Figure 5G have been annotated.

7. Immunofluorescence data in Fig. 6E shows many, if not the majority, of the cells express nuclear N and P. This is unusual.

Reply: Thanks for the comments. We are sorry that N or P and ISG15 were incorrectly marked during image processing due to our mistake. It has been corrected in the newly submitted Fig. 6E.

8. ISG15-dependent inhibition of N-P protein interactions differ significantly between the two assays in Fig. 7E & F. In particular, the luciferase assay produced negligible differences. Does this explain the small differences observed in viral titres?

Reply: Thanks for the comments. We also agree with that since the difference is not obvious during the luciferase competition assay, it may explain the small difference observed in viral titers.

Minor:

The manuscript should be written in past tense.

Line 235-238 - text is confusing.

Reply: Thanks for the comments. The language of the article has been revised.

August 8, 2022

Prof. Guangqing liu
Shanghai Veterinary Research Institute
Chinese Academy of Agricultural Sciences
No.518 Ziyue Road
shanghai
China

Re: Spectrum01031-22R1 (Free ISG15 Inhibits the Replication of Peste des petits Ruminants Virus by Breaking the Interaction of Nucleoprotein and Phosphoprotein)

Dear Prof. Guangqing liu:

Your manuscript has been accepted, and I am forwarding it to the ASM Journals Department for publication. You will be notified when your proofs are ready to be viewed.

Sincerely,

Daniela Rajao
Editor, Microbiology Spectrum

The type I IFN system is at the frontline in defending pathogen invasion. During PRRV infection, IFN- β is substantially induced. In the present study, the author identifies ISG15 as ultimate effector of type I IFN system and characterizes its mechanism of action by perturbing PRRV genomic replication. This study not only facilitates the understanding of ISG15 antiviral strategies, but also provides new clues for developing therapeutic options against PRRV infection.

After careful evaluation of the revised version of manuscript, we find that substantial advances have been made in improving the quality of the manuscript. All the concerns have been fully addressed by the author. The data are more solidated and informative. I believe that this new version of manuscript reaches the requirements for publication in Ms. I highly recommend the editor to accept the manuscript.